# Optimal Effects of Combined Application of Nitrate and Ammonium Nitrogen Fertilizers with a Ratio of 3:1 on Grain Yield and Water Use Efficiency of Maize Sowed in Ridge–Furrow Plastic Film Mulching in Northwest China

**Zhengjun Cui** [1,2], **Yuhong Gao** [1,2,*], **Lizhuo Guo** [1,2], **Bing Wu** [1,3], **Bin Yan** [1,2], **Yifan Wang** [1,2], **Hongsheng Liu** [4], **Gang Li** [5], **Yingze Wang** [1,2] **and Haidi Wang** [1,2]

1   State Key Laboratory of Arid Land Crop Science (SKLALCS), Lanzhou 730070, China
2   College of Agronomy, Gansu Agricultural University (GSAU), Lanzhou 730070, China
3   College of Life Science and Technology, Gansu Agricultural University (GSAU), Lanzhou 730070, China
4   Huining Promotion Center of Agricultural Tecnology (HPCAT), Baiyin 730799, China
5   Baiyin Promotion Center of Agricultural Tecnology (BPCAT), Baiyin 730900, China
*   Correspondence: gaoyh@gsau.edu.cn; Tel./Fax: +86-139-1925-4286

**Abstract:** Improving water use efficiency is essential for the advancement of agricultural production, particularly in arid and semiarid regions. Two-year field experiments were conducted to study the effects of ridge–furrow (RF) and flat planting (FP) plastic film mulching combined with five different nitrogen (N) fertilizers, N1 (KNO$_3$), the nitrate (NO$_3^-$)/ammonium (NH$_4^+$) mixtures with different pure nitrogen ratios N2 (1:1), N3 (1:3), and N4 (3:1), and the control N5 (urea) on maize dry matter accumulation, soil water content, grain yield, water use efficiency (WUE), and N partial factor productivity. Our results showed that RF and N4 were more efficient than FP for increasing maize grain yield, WUE, and nitrogen partial factor productivity, and there was a significant interaction for cultivation practices × N formulation. RF and 3:1 NO$_3^-$/NH$_4^+$ significantly increased grain yield by 14.73% and 13.15%, and 20.07% and 24.14% in 2016 and 2017, respectively, compared to FP and nitrate only. RFN4 produced the highest grain yield in 2016 and 2017 due to the highest dry matter accumulation at filling and physiological maturity stage, ear rows per spike, and row grains per row. Over two growing seasons, the WUE and N partial factor productivity under RFN4 were 18.75% and 29.17% more on average than those of other treatments. Therefore, RFN4 is an effective planting system for increasing the simultaneity of grain yield and WUE for maize production in rain-fed agriculture.

**Keywords:** spring maize; field management practices; nitrogen fertilizers; grain yield; water–nitrogen use efficiency

## 1. Introduction

In recent years, extreme weather such as water scarcity and drought has been becoming increasingly serious, particularly in rain-fed regions in the world [1]. Moreover, global climate change has had an impact on the amount and distribution of precipitation, which has not only decreased crop water use, but also made drought stress increasingly severe, and thus sustainable agricultural production is facing unprecedented challenges [2]. Technologies that enhance crop WUE are vitally important for sustainable crop production and local food security [3]. Plastic film mulch combined with N fertilizer could improve the WUE and properly satisfy the N demands of maize, resulting in an increased grain yield, and realizing the sustainable production of arid and semiarid agriculture [4].

N is an essential macronutrient and limited supplies can limit plant yield and productivity [5]. The element is a constituent of structural biomolecules and is needed for plant physiological metabolism. It can affect photosynthesis directly or indirectly by influencing

the whole process of a plant's carbon and nitrogen metabolism [6]. Nitrate N ($NO_3^-$-N) and ammonium N ($NH_4^+$-N) are major N forms absorbed by plants, which together account for nearly 70% of the total cations and anions; these two forms of N are both effective N sources taken up directly by plants [7]. The composition form of N fertilizer has a significant effect on plant growth and development [8], and the relative contributions of $NO_3^-$-N and $NH_4^+$-N's depend on the types of plant, environmental conditions, developmental stages, and total N concentrations [9].

It was found that plants absorbed more $NO_3^-$ or $NH_4^+$ when this form of soil nitrogen source was in the majority [10]. However, although most plants can use both, $NO_3^-$ is generally the preferred source for crop growth [11]. Excessive $NO_3^-$ can cause negative effects, and high concentrations of $NO_3^-$ lead to $NO_3^-$-N and $NH_4^+$-N accumulation, which causes salt toxicity and affects plant growth [12,13]. The application of different forms of N might cause variations in the absorbed amounts of N, which would then result in an obvious difference in N contents in plant tissues, leading to differences in crop growth and development, and the yield of biomass and grain [7]. Studies have shown that $NO_3^-$ transforms into $NH_4^+$ under the assimilation of nitrogen microorganisms, and when $NH_4^+$ is used as the only supply of N, plant development is suppressed [14]. With identical amounts of N, the application of $NO_3^-$-N fertilizer could produce a greater crop yield than that of $NH_4^+$-N fertilizer [15]. Results indicate that different plants prefer varying amounts of $NO_3^-$-N and $NH_4^+$-N [7]. For example, rice preferred $NH_4^+$ to $NO_3^-$, whereas wheat, tomato, and barley preferred $NO_3^-$ to $NH_4^+$ nutrition [16]. For most plants, fertilizing with a combination of $NO_3^-$ and $NH_4^+$ is superior to solely $NO_3^-$ or $NH_4^+$ fertilization [17].

Maize (*Zea mays* L.) is widely cultivated worldwide for food, fodder, and industrial usage, both in tropical and temperate zones [18]. According to the FAO [19], the world maize planting area in 2020 was 201.84 million hectares, and the area for China was 41.26 million hectares. Maize is one of the primary crops in the rain-fed and drought-prone regions of northern China, accounting for 27.3% of the total arable land area in that part of the country. Due to variations in rainfall, maize yields there vary significantly on a yearly basis [20]. Crop yields and WUE could be improved by soil moisture conservation and reduced evaporation by means of plastic film mulching [21]. RF is an effective measure to maximize precipitation utilization and improve the grain yield and WUE of maize [20]. However, prolonged and extensive use of ridge–furrow plastic film mulching could lead to excessive depletion of soil nutrients and water [22], so improved WUE and nitrogen use efficiency are critical for maize production.

Results showed that the mixed application of $NO_3^-$/$NH_4^+$ had an important effect for improving plant growth and population productivity [10]. It has been reported that $NH_4^+$-N inhibited maize seedling growth, and $NO_3^-$ supply increased the accumulation of maize dry matter at seedling stage [23], while Fan et al. reported that the biomass accumulation in maize was less apparent with $NH_4^+$ supplementation than that with $NO_3^-$ [24]. In addition, different $NO_3^-$/$NH_4^+$ ratios could affect soil water storage in maize cultures at 0–200 cm soil depth, and significantly affect grain yield and WUE [6]. N forms could also affect water uptake and translocation of the element in white clover (*Trifolium repens* L.) and wheat (*Triticum aestivum* L.), whose plants had a higher WUE when they were grown on $NH_4^+$ compared with $NO_3^-$ [25]. Therefore, the grain yield and WUE of maize grown with different mulching types are likely to be affected by the N formula.

Film mulching and the nitrogen formula both affect maize growth, grain yield, productivity, and WUE [26,27]. Film mulching modifies soil water utilization by lowering the evaporation-to-transpiration ratio. An appropriate supply of $NH_4^+$ in addition to nitrate can significantly enhance maize growth and boost production [28]. The independent effects of the mixed application of $NO_3^-$/$NH_4^+$ on maize growth and grain yield have been well documented [26–28]. However, the interactions between film mulching methods and N forms have received less attention, particularly during years with seasonal variations in precipitation. Therefore, we investigated whether the productivity of spring maize in a semiarid region was affected by using RF in combination with various nitrogen fertilizer

formulations. The study's objectives were as follows: (1) to determine the effects of two film mulching methods combined with five nitrogen fertilizers on growth, soil moisture spatial-temporal change, and dry matter accumulation of rain-fed maize cultivated in the dry-land of northern China; (2) to analyze the impacts of nitrogen fertilizers and film mulching techniques on the productivity of maize, and to clarify variations in grain yield, WUE, and N partial factor productivity under various treatments; and (3) to investigate the best combination of the film mulching method and N form on the maize hybrid "Jinping 618" in these regions.

## 2. Materials and Methods

### 2.1. Experimental Site

The experiment was conducted out at Huishi Town, Huining County, Gansu Province, China (105°02′ E, 35°38′ N), during 2016 and 2017. The experimental site is located at an elevation of 1772 m. The local semiarid climate has an annual mean temperature of 8.3 °C, a frost-free period of 155 days, and a mean annual rainfall of 356.70 mm, with a mean annual evaporation of 1800 mm. Figure 1 shows two years' worth of meteorological data collected at the Huining meteorological observatory. The year 2016 saw 251.9 mm of rainfall in total, while 2017 saw 432 mm. The mean maximum air temperature in 2016 varied between 15.3 and 33.3 °C, while the mean minimum air temperature ranged between −2.8 and 17.0 °C. These temperature ranges were 17.1–34.6 °C and −2.3–14.0 °C, respectively, in 2017.

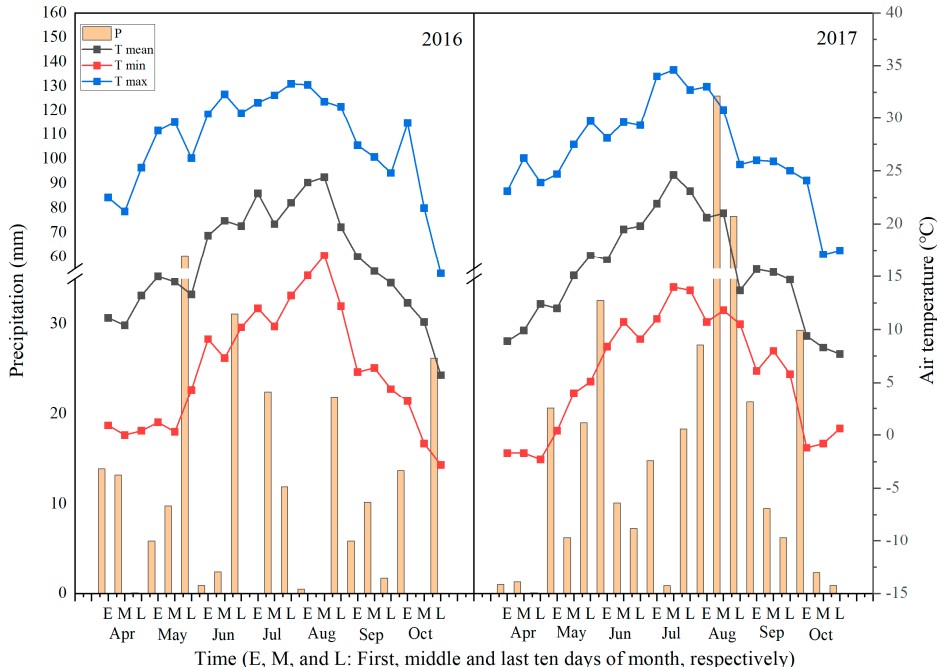

**Figure 1.** Ten-days (the first, middle, and last ten days) precipitation from April to October at the experimental site (P, in mm), mean air temperature (T mean), maximum (T max) and minimum air temperature (T min, all in °C) from April to October, in 2016 and 2017.

### 2.2. Soil of the Experimental Site

The experimental station's local soil was a sandy loam with an average bulk density of 1.23 g cm$^{-3}$ (determined in 2016) and 1.24 g cm$^{-3}$ (determined in 2017). Table 1 provides a summary of other physical and chemical characteristics of the soil layer at a depth of 0–30 cm.

**Table 1.** Soil porosity and chemical characteristics of the soil of the experimental site.

| Year | Soil Porosity (%) | Organic Matter (g kg$^{-1}$) | Total N (g N kg$^{-1}$) | Available P (mg P kg$^{-1}$) | Available K (mg K kg$^{-1}$) | pH |
|---|---|---|---|---|---|---|
| 2016 | 54.3 | 11.96 | 0.98 | 59.87 | 5.46 | 8.17 |
| 2017 | 53.9 | 11.94 | 0.97 | 60.07 | 5.38 | 8.12 |

*2.3. Experimental Treatments and Methodology*

Ten combinations of treatments were included in the experiment. Two different mulching practices were compared: ridge–furrow plastic film mulching (RF) and flat planting with plastic film mulching (FP). These were combined with five different N fertilizers (Table 2): $NO_3^-$:$NH_4^+$ ratios of 1:0 (along $NO_3^-$ N1), 1:1 (N2), 1:3 (N3), and 3:1 (N4), and a control, N5, for which urea (Gansu Liuhua (Group) Co., Linxia, China) was the sole nitrogen source (N content 46%). Irrespective of the N form, all treatments received 300 kg N ha$^{-1}$. The N fertilizers were applied in three splits, with 60% applied as basal N before sowing, 20% applied at jointing stage (BBCH-36), and 20% applied at filling stage (BBCH-75). 3,4-Dimethylpyrazole phosphate (DMPP) is a new nitrification inhibitor with highly favorable properties, and the application rates of 0.5–1.5 kg ha$^{-1}$ DMPP are sufficient to achieve optimal nitrification inhibition [29]. The nitrification inhibitor DMPP (Hubei Shuangyan Chemical Co., Wuhan, China) was applied in N2, N3, and N4 to avoid conversion of $NH_4^+$ to $NO_3^-$ during the experiment; the application rate of DMPP was calculated as 1% of the pure N in $NH_4^+$-N, and DMPP was applied at the same time (before sowing, jointing, and filling stage) as $NH_4^+$-N fertilizer, the application rates were N2 (0.9 kg ha$^{-1}$, 0.3 kg ha$^{-1}$, 0.3 kg ha$^{-1}$), N3 (1.35 kg ha$^{-1}$, 0.45 kg ha$^{-1}$, 0.45 kg ha$^{-1}$), N4 (0.45 kg ha$^{-1}$, 0.15 kg ha$^{-1}$, 0.15 kg ha$^{-1}$), respectively, and the total application rates are shown in Table 2. Additionally, 180 kg of 16% $CaP_2O_5$ ha$^{-1}$ fertilizer (Gansu Baiyin Hubao Chemical Co., Baiyin, China) was applied. In all cases, total K fertilizer was kept constant by addition of potassium sulfate if necessary. $K_2SO_4$, $KNO_3$, and $(NH_4)_2SO_4$ were made by Gansu Liuhua (Group) Co., Linxia, China.

**Table 2.** Details of the treatments.

| Treatment | Symbol | Fertilizer Amount | | DMPP (kg ha$^{-1}$) ‡ |
|---|---|---|---|---|
| | | KNO$_3$ (kg ha$^{-1}$) | (NH$_4$)$_2$SO$_4$ (kg ha$^{-1}$) | |
| Ridge–furrow plastic film mulching | RF | n.a. | n.a. | n.a. |
| Flat planting with plastic film mulching | FP | n.a. | n.a. | n.a. |
| KNO$_3$ fertilizer | N1 | 2221.5 | 0 | 0 |
| KNO$_3$:(NH$_4$)$_2$SO$_4$ with ratio † 1:1 | N2 | 1111.5 | 714 | 1.5 |
| KNO$_3$:(NH$_4$)$_2$SO$_4$ with ratio † 1:3 | N3 | 555 | 1071 | 2.25 |
| KNO$_3$:(NH$_4$)$_2$SO$_4$ with ratio † 3:1 | N4 | 1666.5 | 357 | 0.75 |
| Urea fertilizer | N5 | 646.5 kg ha$^{-1}$ Urea | | 0 |

The total amount of N was kept constant. † ratios are given as $NO_3^-$/$NH_4^+$ and refer to pure N content, ‡ the application of DMPP (3,4-Dimethylpyrazole phosphate) was calculated as 1% of pure nitrogen in $NH_4^+$-N content of the basic fertilizer. n.a. not applicable.

Each plot size was 40.26 m$^2$ (6.1 × 6.6 m). For RF, the plots consisted of 15 cm high, narrow ridges (width 40 cm), alternated with wide ridges (10 cm high, 70 cm wide) (Figures 2 and 3). All ridges were covered with transparent plastic polyethylene film of 0.01 mm thickness that was 140 cm wide (Lanzhou Green Garden Corp., Lanzhou, China). This ensured that the soil temperature rose, and the amount of rainwater collected increased, and the evaporative losses were reduced [30]. For the flat application, the plant spacing was the same. After the soil was covered with film, holes (1.5 cm in diameter) were made using a handheld device in furrows at 35 cm intervals for convenient collection and channeling of precipitation. The plant spacing in each row was 35 cm and kept a full stand of seedlings of 67,500 plants per hectare. The experiment was a two-factor randomized block design

with three replications. There were no diseases or pests that occurred affecting the maize during the complete experiment. The maize (variety Jinping 608, Wuwei Golden Apple Agriculture Co., Wuwei, China) was sown on 21 April 2016, and on 24 April 2017, and crops were harvested on 22 September 2016, and on 8 October 2017.

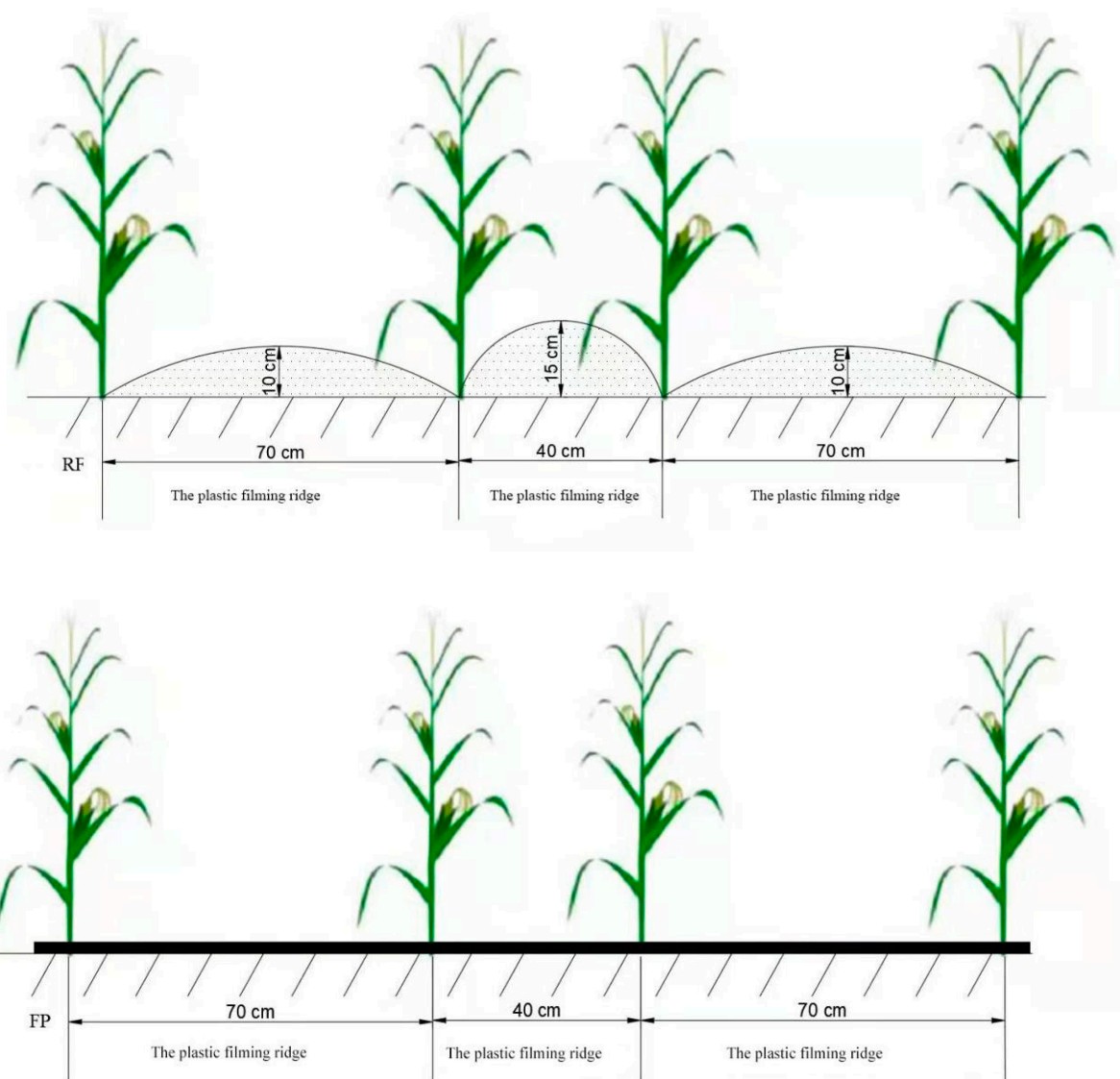

**Figure 2.** Schematic diagram of the field layout where maize plants were separated by alternating distance. The top shows RF (ridge–furrow plastic film mulching) with wide (70 cm) and narrow (40 cm) plastic film-mulched ridges of 10 cm and 15 cm height, respectively. The bottom shows FP, (flat planting with plastic film mulching) with the same plant distance.

*2.4. Growth Parameters and Dry Matter Accumulation*

The maize plants were observed at five developmental stages based on the BBCH scale: seedling (BBCH-13), jointing (BBCH-36), tasseling (BBCH-59), filling (BBCH-75), and physiological maturity (BBCH-87). At each stage, three plants were chosen at random from each plot. After determining the total leaf area, the leaves' length and width were measured, and the leaf area index was calculated as follows [31]:

$$\text{LAI} = \text{the total leave area/occupied land area} \tag{1}$$

At the tasseling stage, filling stage, and maturity stage, three plants were randomly selected from each plot and leaves, stems and ears were placed into an oven at 105 °C

for 30 mins and then oven-dried to a constant weight at 75 °C [32]. The dry matter accumulation per hectare was calculated from the average weight of three plants multiplied by the plants per hectare.

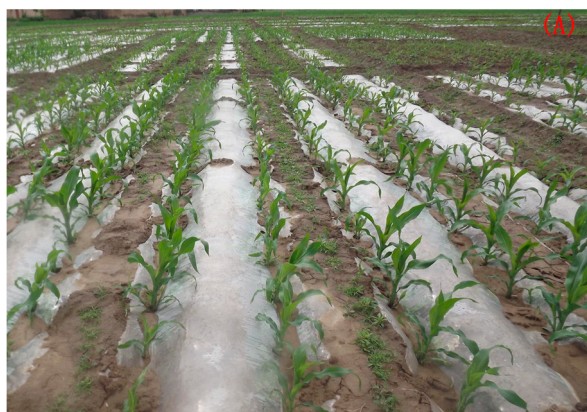 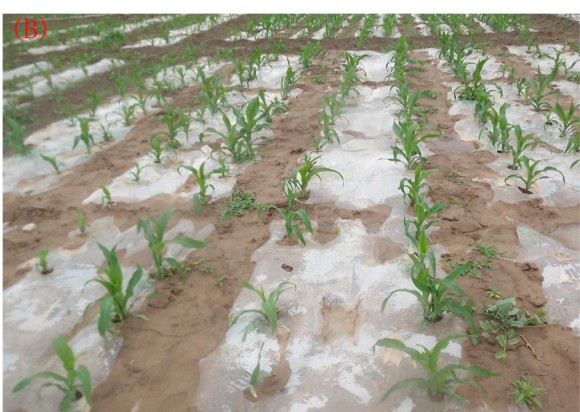

**Figure 3.** Experimental (**A**) ridge–furrow plastic film mulching and (**B**) flat planting with plastic film mulching.

### 2.5. Soil Water Calculations

Soil was collected at a depth of 0–200 cm using a soil auger (4.5 cm diameter) at the different maize growth stages as previously described [33] and gravimetric soil water content was measured at 20 cm depth intervals. For this, from each plot, three continuous furrows were randomly selected. The collected soil samples were placed in an aluminum box, weighed immediately, dried at 105 °C until constant weight, and then weighed again. SWC was calculated using the following formula:

$$\text{SWC (\%)} = (\text{FW} - \text{DW})/(\text{DW} - \text{AW}) \times 100 \qquad (2)$$

where FW is fresh weight of the soil sample with aluminum box, DW is dry weight of soil sample with aluminum box, and AW is the weight of aluminum box. The soil water storage (SWS, in mm) was calculated as follows [34]:

$$\text{SWS} = \text{h} \times \rho \times \theta \times 10^{-1} \qquad (3)$$

where h is the soil depth (cm), $\rho$ is the soil bulk density (g cm$^{-3}$), and $\theta$ is the SWC (%).

### 2.6. Crop Water Use

Evapotranspiration (ETa), which was calculated using the soil water balance equation, was used to characterize the crop season's water usage or consumptive use of water under the various treatments.

$$\text{ETa} = \text{I} + \text{P} - \text{R} - \text{D} \pm \Delta\text{SWS} \qquad (4)$$

where I is the amount of irrigation (mm), which was zero in this experiment, P is precipitation (mm), R is the surface runoff (mm), D is the deep seepage (mm), and $\Delta$SWS is the change in soil water storage (mm), which is the difference between soil water content at planting and at harvesting [35]. According to the actual conditions, the contributions of groundwater recharge, runoff, and deep seepage were negligible during the experiments.

### 2.7. Grain Yield, Harvest Index, and Water Use Efficiency

At physiological maturity, every maize plant was physically removed from a plot that was 20 m$^2$ (4 m × 5 m) in size. For each plot, the weight of the air-dried grain and grain yield were determined. The measurements of the 100-grain weight, spike length, ear breadth, ear barren tip, ear rows, row grains, and PFP were taken. The harvest index was a

measure of the grain yield to dry matter accumulation above ground [36]. The WUE for grain yield and biomass yield was determined as shown below.

$$WUEb = BY/ETa \tag{5}$$

$$WUEg = GY/ETa \tag{6}$$

where BY is biomass yield (kg ha$^{-1}$), GY is grain yield, and ETa is the crop water consumption. WUEb is water use efficiency of biomass yield (kg ha$^{-1}$ mm$^{-1}$), and WUEg is water use efficiency of grain yield (kg ha$^{-1}$ mm$^{-1}$) [33,37].

### 2.8. Statistical Analysis

The value of each indicator was the mean of three replicates per treatment, and the SPSS 20.0 software (SPSS Inc., Chicago, IL, USA) was used to perform the analysis of variance. Data preprocessing was performed using Excel 2016. The figures were plotted using Origin 2019. All pairwise comparisons of the treatment means were performed using the least significant difference (LSD) test with significance determined at the 5% level.

## 3. Results

### 3.1. Growth of Maize Plants under Different Nitrogen Fertilizer and Cultivation Practices

The plant heights and stem diameters were significantly impacted by cultivation practices (CP) and N formula treatments, and plant height had a significant interaction with CP × N (Table 3). Under the same cultivation practice conditions, plant height and leaf area index were significantly higher for the $NO_3^-$:$NH_4^+$ = 3:1 (N4) than the control (N5) ($p < 0.05$). Stem diameter was significantly higher in the N4 than the $NO_3^-$:$NH_4^+$ = 1:0 (N1), $NO_3^-$:$NH_4^+$ = 1:1 (N2), and $NO_3^-$:$NH_4^+$ = 1:3 (N3) treatments ($p < 0.05$), which indicates that stem diameter was positively correlated with the amount of nitrate (Table 3). This indicates that $NO_3^-$:$NH_4^+$ = 3:1 was more conducive to increases in the plant height, stem diameter, and leaf area index of maize. The change trends for plant height and stem diameter under the same N fertilizer conditions were similar, and the stem diameter in ridge–furrow plastic film mulching was higher than flat planting with plastic film mulching. In general, the RFN4 treatment had the tallest plants with heights of 197.99 and 208.44 cm and stem diameters of 31.06 and 30.78 mm, whereas the RFN2 and RFN4 treatments had the highest leaf area indices of 6.07 and 6.02, respectively, in 2016 and 2017.

**Table 3.** Effect of culture practice and nitrogen fertilizer on plant height (cm), stem diameter (mm), and LAI at harvest of spring maize.

| Treatments | | Plant Height (cm) | | Stem Diameter (mm) | | LAI | |
|---|---|---|---|---|---|---|---|
| | | 2016 | 2017 | 2016 | 2017 | 2016 | 2017 |
| | N1 | 187.26 [b] | 191.00 [b] | 29.50 [b] | 28.36 [c] | 5.91 [ab] | 5.86 [ab] |
| | N2 | 195.64 [a] | 200.78 [ab] | 30.02 [ab] | 29.54 [b] | 6.07 [a] | 5.99 [a] |
| RF | N3 | 196.25 [a] | 192.44 [b] | 29.88 [b] | 29.32 [b] | 5.75 [b] | 5.70 [b] |
| | N4 | 197.99 [a] | 208.44 [a] | 31.06 [a] | 30.78 [a] | 6.02 [a] | 6.02 [a] |
| | N5 | 190.55 [b] | 175.22 [c] | 29.24 [b] | 29.06 [b] | 5.62 [b] | 5.36 [b] |
| | N1 | 185.67 [ab] | 184.67 [ab] | 28.85 [b] | 27.92 [b] | 5.83 [ab] | 5.71 [ab] |
| | N2 | 179.26 [b] | 181.33 [b] | 28.66 [b] | 28.11 [b] | 5.70 [ab] | 5.66 [ab] |
| FP | N3 | 188.97 [ab] | 182.22 [b] | 28.78 [b] | 28.77 [b] | 5.49 [b] | 5.49 [b] |
| | N4 | 192.34 [a] | 189.22 [a] | 29.86 [a] | 29.72 [a] | 5.97 [a] | 5.91 [a] |
| | N5 | 182.33 [b] | 171.44 [c] | 28.55 [b] | 28.23 [b] | 5.34 [b] | 5.20 [b] |
| CP | | ** | * | * | * | ns | ns |
| N | | ** | ** | * | * | * | * |
| CP × N | | * | * | ns | ns | ns | ns |

LAI, leaf area index; CP, cultivation practice (FP or RF); N1, $NO3^-$:$NH4^+$ ratios of 1:0; N2, $NO3^-$:$NH4^+$ ratios of 1:1; N3, $NO3^-$:$NH4^+$ ratios of 1:3; N4, $NO3^-$:$NH4^+$ ratios of 3:1; N5, urea. N, nitrogen formula. * means a significant difference ($p < 0.05$); ** means a very significant difference ($p < 0.01$); and ns means no significant difference ($p > 0.05$). Different lowercase letters after the values in the same column indicated that the difference was significant at $p < 0.05$ under LSD test.

### 3.2. Dry Matter Accumulation

Maize dry matter accumulation was impacted by cultivation practices and N formulas in the analyzed developmental stages, with significant interactions for CP × N in all stages except for the seedling stage (Table 4). Compared with FP, dry matter accumulation at jointing, filling, and physiological maturity stages were significantly higher under RF treatments in both years. This might be because RF could collect runoff from ridges, improve soil water storage, prolong the period of maze water availability, and enhance maize growth. When compared to the N1 and N5 treatments, the dry matter accumulation under the N4 treatment was much higher during the grain filling and physiological maturity stages (Figure 4). When a mixture of $NO_3^-$-N and $NH_4^+$-N was provided in the growth medium, especially in arid and semiarid environments, the forms of the applied N had a substantial impact on plant growth and development. Many crops had higher dry matter accumulation than the $NO_3^-$-fed or $NH_4^+$-fed plants.

**Table 4.** Significance of the effects of treatments on dry matter accumulation at various stages of plant development.

| Treatment | Seedling Stage | | Jointing Stage | | Tasseling Stage | | Filling Stage | | Physiological Maturity Stage | |
|---|---|---|---|---|---|---|---|---|---|---|
| | 2016 | 2017 | 2016 | 2017 | 2016 | 2017 | 2016 | 2017 | 2016 | 2017 |
| CP | * | *** | ** | ns | ** | *** | *** | *** | *** | *** |
| N | ns | ns | ** | *** | *** | *** | *** | *** | *** | *** |
| CP × N | ns | ns | *** | *** | *** | ** | * | ** | ** | *** |

CP, cultivation practice (RF or FP), N, nitrogen formula. Significance is indicated as * for $p < 0.05$; ** for $p < 0.01$, *** for $p < 0.001$; and ns, not significant ($p > 0.05$).

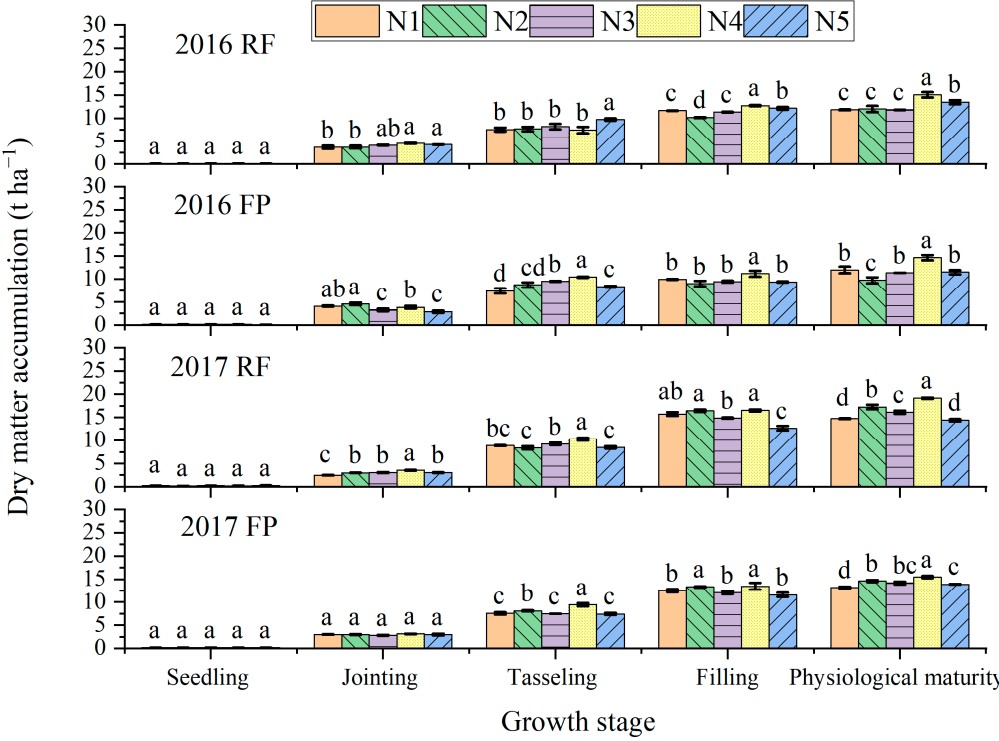

**Figure 4.** Effects of different cultivation practices and nitrogen formulas on dry matter accumulation in 2016 and 2017. RF, ridge–furrow plastic film mulching; FP, flat planting with plastic film mulching; N1, $NO_3^-$:$NH_4^+$ ratios of 1:0; N2, $NO_3^-$:$NH_4^+$ ratios of 1:1; N3, $NO_3^-$:$NH_4^+$ ratios of 1:3; N4, $NO_3^-$:$NH_4^+$ ratios of 3:1; N5, urea. Bars show mean values plus one standard error ($n = 3$). Different lowercase letters above the bars indicate that the difference was significant at $p < 0.05$ under LSD test.

The formulas of N did not significantly influence the dry matter accumulation at seedling stage; this was due to the low accumulation of dry matter and the differences among treatments had not yet emerged. At jointing stage, the dry matter accumulation of N4 was the highest under the RF practice, which was significantly increased compared to N1. In the mixed N formulas, the significantly highest dry matter accumulation was found in the N2 treatment under the FP practice in 2016. Likewise, at the tasseling stage, dry matter accumulation was the highest with the N4 treatment (except for RF in 2016). At the filling and maturity stages, the highest dry matter accumulation was found in the treatment of N4, no matter whether under RF or FP. Variations in the N forms may have affected the water relations of plants in different ways for the N nutrition, which then affected the plant growth and dry matter accumulation.

*3.3. Soil Water Content*

The soil water dynamics during the different growth stages of the plants in 2016 and 2017 are shown in Figure 5. The soil water content determined at various depths varied with the year. The water content of the soil also varied with the different N formulas, with similar trends for the two different cultivation practices in the same year.

At the seedling stage, the soil water content of the soil layer at 40–80 cm was significantly higher in FP than in RF in both years. In 2016, the water content of the soil at 0–40 cm depth in the N4 treatment was reduced significantly compared with N5, for both RF and FP (Figure 5A). In contrast, the N4 treatment significantly increased (by 13.7%) the soil water content at the same depth under RF in 2017 (Figure 5A). On average, the largest water content of the soil of maize was recorded from RFN5 (19.68%) in 2016 and RFN1 (9.78%) in 2017, respectively. This might be related to the rainfall, as the rainfall at the seedling stage in 2016 was much lower than that in 2017.

At the jointing stage, the average soil water content of the 0–200 cm soil layer was lower in 2016 than that in 2017, the water content of the soil in FP was significantly higher than that under the RF practice, and the N4 treatment had the highest water content of the soil (Figure 5B). Under the RF practice, N4 had an average soil water content that was 5.59 percent and 9.09 percent higher than N5 at depths of 0–20 and 40–80 cm, respectively. In 2016, RFN3 had the greatest average soil water content, but in 2017, RFN5 had the highest average soil water content.

Compared with FP, RF significantly improved the water content of the soil of the 20–100 cm soil layer in 2016. N5 had the highest soil water content at 0–20 cm in 2016 (Figure 5C), which was approximately 17.59% higher (RF) and 8.43% higher (FP) than N4. There was no significant difference in the water content of the soil of 0–20 cm between N2 and N4 treatments under RF in 2017, which both produced higher values than N1, N3, and N5. It was indicated that RF cultivation practices were better than FP in collecting rainfall and reserving moisture, and N4 was more beneficial for improving soil water content during the tasseling stage in dry years.

At the filling stage, the soil water content at a depth of 0–20 cm was higher under the RF than the FP practice in 2016, whereas in 2017, RF had lower water content of the soil values than FP. This interannual inconsistency was most likely related to the extreme drought during the filling stage in 2017, whereby plants grown under RF tended to consume more water as they produced more aboveground biomass (Figure 5D). The highest average soil water content of the 0–200 cm soil layer was recorded for the N5 treatment, although the N formula had no significant effect on it.

During the physiological maturity stage, the water content of the soil at soil depths of 0–20 cm and 40–100 cm was higher under FP than it was in RF, indicating that RF led to increased water use during the growing season. For both farming techniques in 2016, N4 considerably decreased the water content of the soil in the 0–20 cm soil layer at the maturity stage (Figure 5E). Due to the severe drought, there was no discernible difference in 2017 between the cultivation methods and N formula in terms of soil water content.

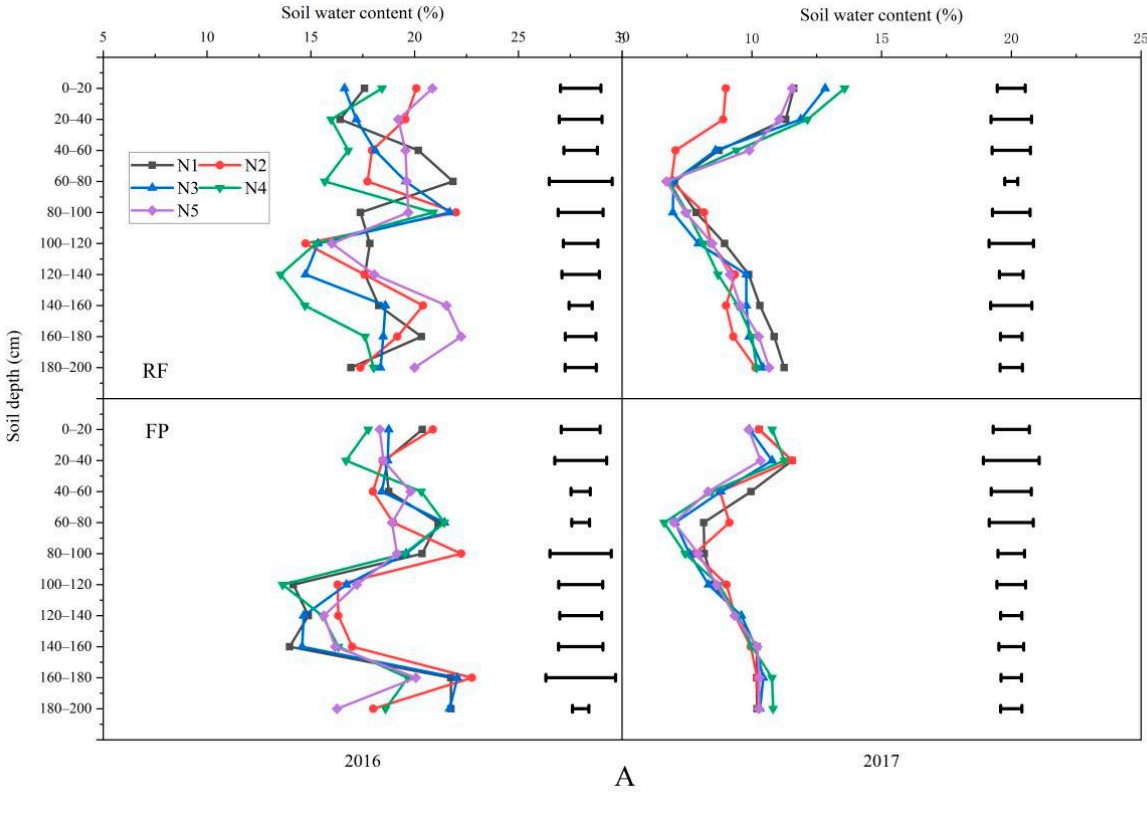

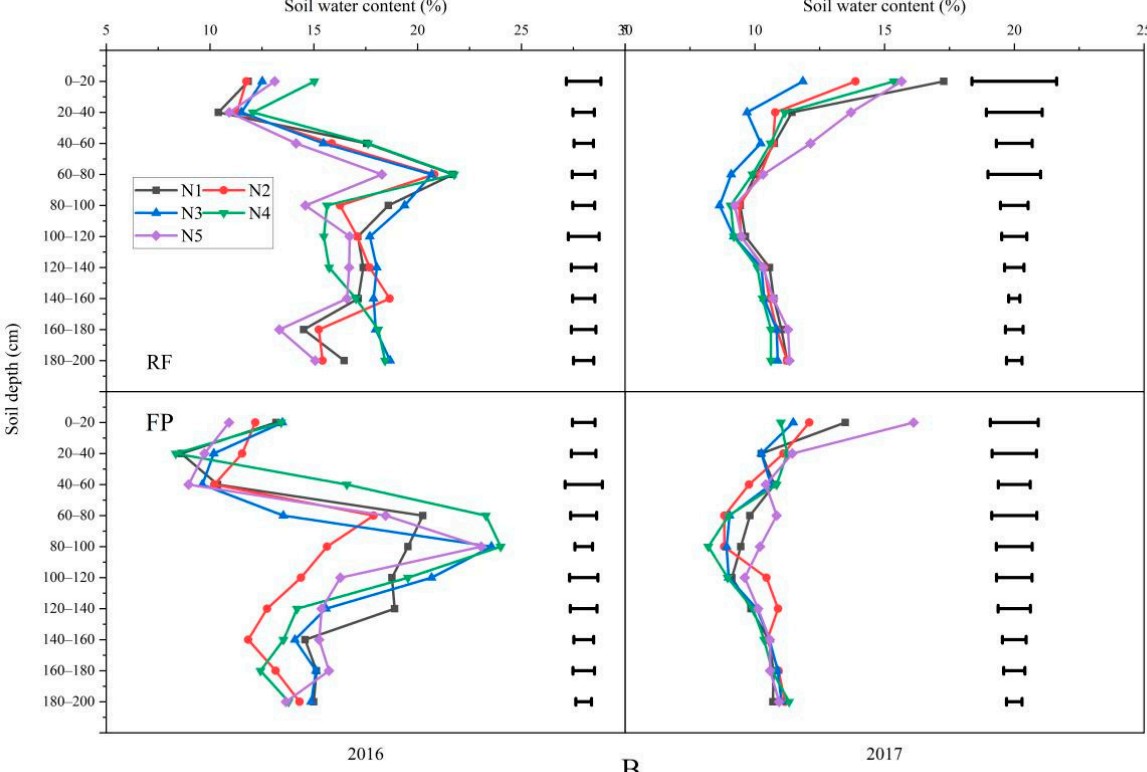

**Figure 5.** *Cont.*

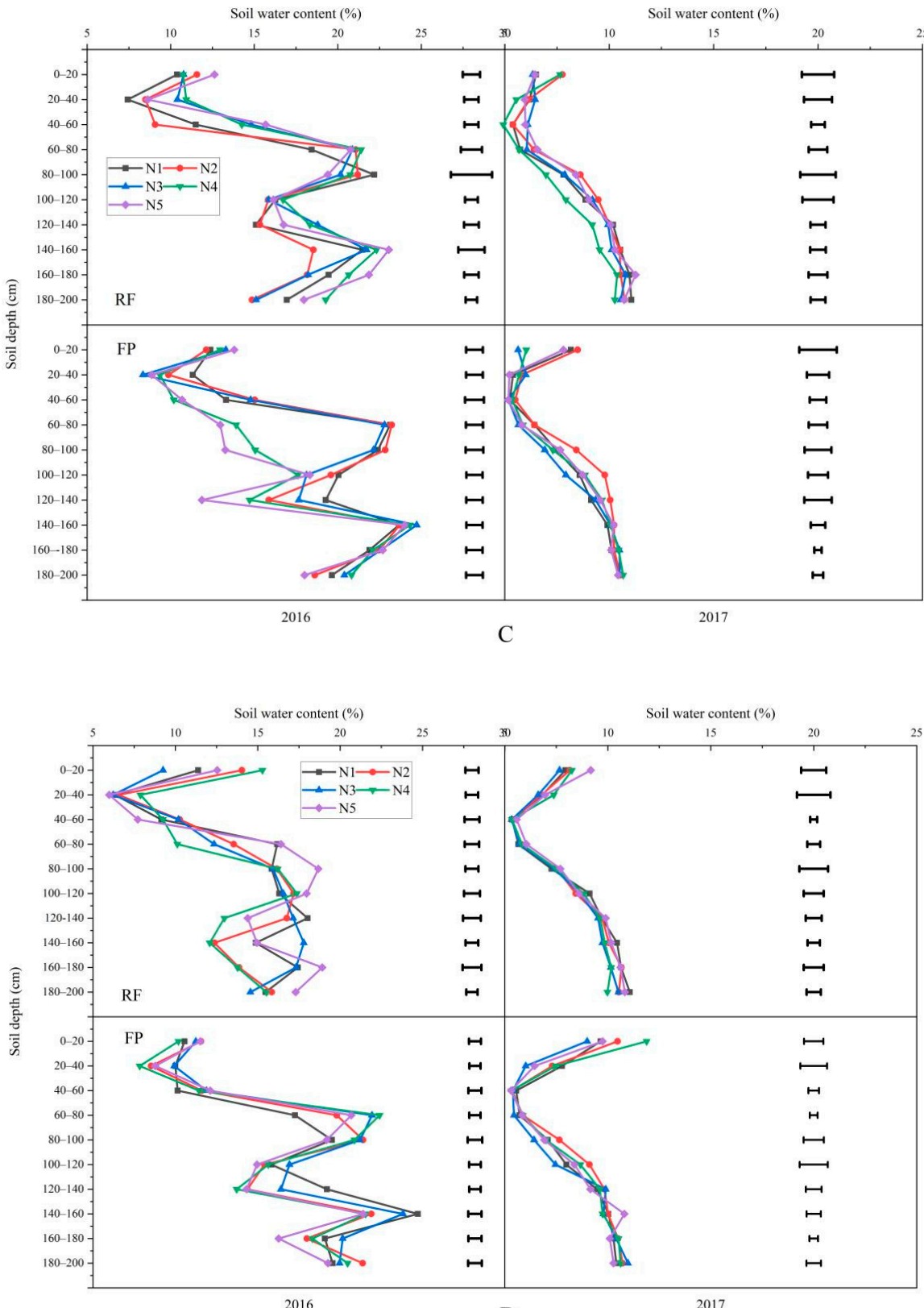

**Figure 5.** *Cont.*

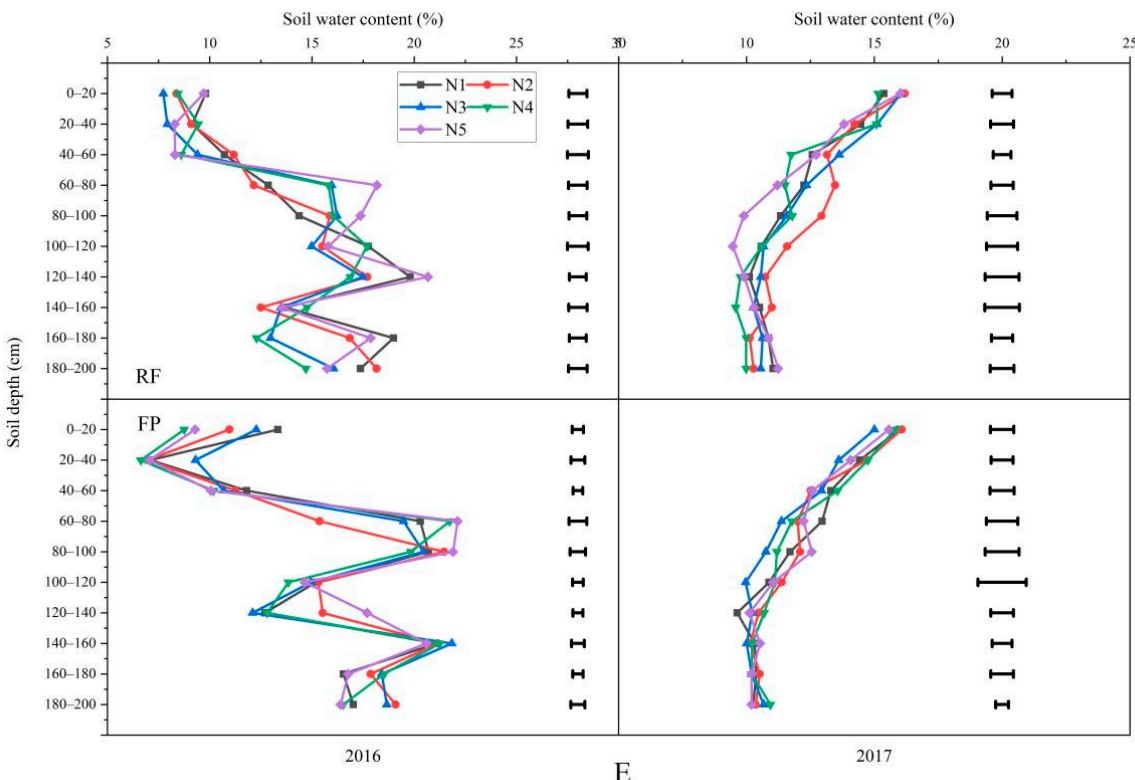

**Figure 5.** Soil water content at variable depths up to 200 cm during the growth of the plants, recorded for 5 developmental stages in 2 years. Four panels are shown per stage, with (clockwise starting top left) RF in 2016, RF in 2017, FP in 2017, and FP in 2016. (**A**) Seedling stage; (**B**) jointing stage; (**C**) tasseling stage; (**D**) filling stage; (**E**) physiological maturity stage. RF ridge–furrow plastic film mulching; FP flat planting with plastic film mulching. N1, $NO_3^-$:$NH_4^+$ ratios of 1:0; N2, $NO_3^-$:$NH_4^+$ ratios of 1:1; N3, $NO_3^-$:$NH_4^+$ ratios of 1:3; N4, $NO_3^-$:$NH_4^+$ ratios of 3:1; N5, urea.

### 3.4. Evapotranspiration

The evapotranspiration (ETa) varied with the year and was positively affected by different cultivation practices and nitrogen formula (Figure 6). Compared with 2016, ETa was significantly higher in 2017, which was associated with the little rainfall in this year (Figure 1). The cultivation practices had a significant influence on ETa in 2016, due to the fact that extreme drought exacerbated ETa, resulting in no difference in ETa between the RF and FP treatments in 2017. The ETa was affected by the N formula; compared with N5, the ETa of N4 increased by 9.74% and 6.45% under the RF and FP treatments, respectively, in 2016, and there was no significant difference in 2017. The ETa was significantly higher for the RFN3 treatment and for the PFN4 treatment in 2016 (Figure 6). This suggested that the effects of the cultivation practices and N formula on ETa were related to rainfall during the maize growth period, and the effects of the cultivation practices and N formula showed no significant differences in ETa, such as in 2017, when rainfall was less than 252 mm.

### 3.5. Yield Components

A combined ANOVA for the related traits of spike length, ear width, ear barren tip, ear rows, row grains, and one-hundred (100)-grain weight revealed that there was a significant cultivation practice and N formula effect for almost all traits (Table 5). A significant CP × N treatment interaction was detected for spike length, ear width, ear rows, row grains, and one-hundred-grain weight (Table 5).

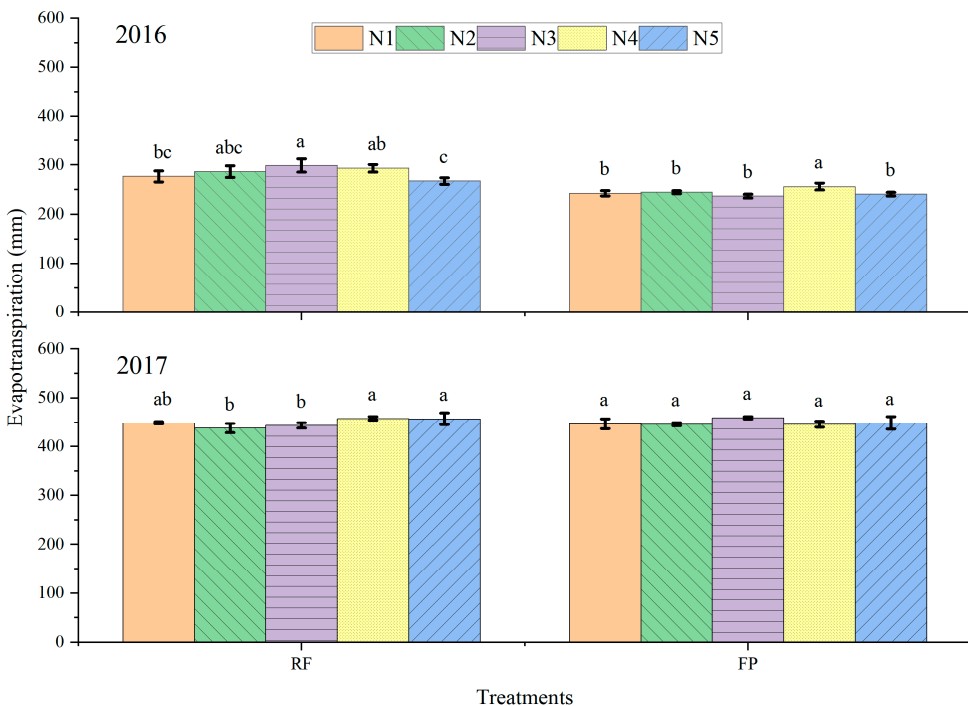

**Figure 6.** Effects of different treatments on evapotranspiration (ETa). N1, $NO_3^-:NH_4^+$ ratios of 1:0; N2, $NO_3^-:NH_4^+$ ratios of 1:1; N3, $NO_3^-:NH_4^+$ ratios of 1:3; N4, $NO_3^-:NH_4^+$ ratios of 3:1; N5, urea. Bars showed mean values plus one standard error (*n* = 3). Different letters above the bars indicated that a significant difference was at *p* < 0.05 according to an LSD test.

**Table 5.** Effects of different treatments on various yield components of the harvested maize.

| Year (Y) | Treatment | | Spike Length (cm) | Ear Width (mm) | Ear Barren Tip (cm) | Ear Rows (rows) | Row Grains (grains) | 100-Grain Weight (g) |
|---|---|---|---|---|---|---|---|---|
| 2016 | RF | N1 | 14.24 c | 43.33 a | 2.17 a | 16.00 b | 27.11 c | 19.13 d |
| | | N2 | 15.03 b | 42.44 ab | 1.33 b | 16.89 a | 28.44 bc | 20.40 c |
| | | N3 | 15.67 a | 41.77 b | 1.07 c | 16.00 b | 30.89 a | 21.79 b |
| | | N4 | 15.63 a | 43.27 a | 0.83 d | 17.00 a | 31.22 a | 24.44 a |
| | | N5 | 14.83 b | 41.71 b | 1.42 b | 16.22 b | 30.22 ab | 21.76 b |
| | FP | N1 | 14.80 ab | 40.83 b | 1.25 ab | 15.67 c | 25.56 d | 18.90 a |
| | | N2 | 15.00 a | 39.64 b | 1.58 a | 16.90 a | 27.11 c | 19.54 a |
| | | N3 | 15.02 a | 40.66 b | 1.50 ab | 16.44 ab | 28.67 b | 20.35 a |
| | | N4 | 15.42 a | 42.47 a | 1.17 b | 17.00 a | 30.44 a | 20.41 a |
| | | N5 | 14.28 b | 40.60 b | 1.17 b | 16.00 bc | 30.33 a | 21.24 a |
| 2017 | RF | N1 | 14.80 a | 44.73 c | 0.73 b | 13.38 c | 26.00 b | 36.94 a |
| | | N2 | 16.50 a | 47.39 b | 1.02 ab | 18.17 b | 29.33 ab | 36.28 a |
| | | N3 | 16.75 a | 44.87 c | 0.84 b | 17.13 b | 27.63 b | 36.32 a |
| | | N4 | 17.16 a | 52.45 a | 0.80 b | 21.33 a | 32.67 a | 38.31 a |
| | | N5 | 15.89 a | 39.55 d | 1.29 a | 12.92 c | 19.50 c | 34.95 a |
| | FP | N1 | 12.33 d | 39.73 c | 1.05 ab | 14.13 b | 18.75 c | 33.87 a |
| | | N2 | 15.48 ab | 43.33 b | 1.19 ab | 15.25 ab | 21.75 b | 36.47 a |
| | | N3 | 15.19 b | 38.34 d | 1.60 a | 14.63 b | 18.58 c | 36.58 a |
| | | N4 | 16.00 a | 45.41 a | 0.75 b | 16.33 a | 26.25 a | 36.80 a |
| | | N5 | 14.31 c | 40.13 c | 1.58 a | 12.08 c | 17.58 c | 35.79 a |
| CP | | | *** | *** | * | *** | *** | * |
| N | | | *** | *** | *** | *** | *** | ** |
| CP × N | | | *** | ** | *** | * | * | ns |

Note: Y, year, CP, cultivation practice (FP or RF); N, nitrogen formula; significance indicated as * for *p* < 0.05; ** for *p* < 0.01, *** for *p* < 0.001; and ns, not significant (*p* > 0.05). Different lowercase letters after the values in the same column indicate that the difference was significant at *p* < 0.05 under LSD test.

Regarding the yield components, the spike length, ear width, ear rows, row grains, and one-hundred-grain weight were significantly decreased, whereas the ear barren tip was higher under the FP treatment. The RF helped to efficiently channel rainwater toward furrows, reduced soil evaporation throughout maize growth, and increased soil water storage, improving maize components. There were significant differences for ear width and ear rows between the N4 and N5 treatments; the ear width and ear rows of N4 were higher than N5 no matter whether under RF or FP treatment in 2016 and 2017. This suggested that a mixed supply of the two forms of N stimulated the growth of maize, and improved the ear width and ear rows. On average, RFN4 showed higher spike length, ear width, row grains, and one-hundred-grain weight than the other N treatments under both cultivation practice treatments.

### 3.6. Grain Yield, Harvest Index, Water Use Efficiency, and Partial Factor Productivity

Cultivation methods, nitrogen formula, and substantial CP N interactions all significantly influenced maize grain yield, WUE, and nitrogen partial factor productivity (Table 6). The mean grain yields in the RF treatment were 5869.41 kg ha$^{-1}$ and 4970.67 kg ha$^{-1}$ in 2016 and 2017, and 5116.00 kg ha$^{-1}$ and 4392.99 kg ha$^{-1}$ in the FP treatment, respectively (Table 6). In comparison to FP, the RF cultivation method enhanced the WUEb by 8.35% in 2016 and 14.60% in 2017, as well as the productivity of the N partial factor by 14.74% in 2016 and 13.11% in 2017. In 2017, maize WUEg under RF grew dramatically by 13.18 percent; however, in 2016 there was no discernible difference. N4 performed much better than N5 among the nitrogen treatments in terms of grain yield, biomass WUE, grain WUE, and N partial factor productivity. N4 significantly increased nitrogen partial factor productivity by 28.18% in 2016 and 38.34% in 2017 compared to the N5 treatment, increasing WUEb by 19.20% in 2016 and 22.39% in 2017, WUEg by 28.17% in 2016 and 38.31% in 2017, and grain yield by 27.90% in 2016 and 38.04% in 2017. Among all the treatments, maximum grain yield, WUE, and nitrogen partial factor productivity were found in the treatment of RFN4, with values 6779.28 kg ha$^{-1}$, 32.90 ka ha$^{-1}$ mm$^{-1}$, 23.17 kg ha$^{-1}$ mm$^{-1}$, and 37.66 kg kg$^{-1}$ in 2016 and 5990.22 kg ha$^{-1}$, 41.86 ka ha$^{-1}$ mm$^{-1}$, 13.08 kg ha$^{-1}$ mm$^{-1}$, and 33.28 kg kg$^{-1}$ in 2017 (Table 6). These values were higher than the average values of the region. The results indicate that FP was the optimal cultivation practice for better performance of grain yield, WUE, and N partial factor productivity, and N4 was the optimal N formula for better performance for grain yield, WUE, and N partial factor productivity.

### 3.7. Correlation Analysis

The dry matter accumulation was positively correlated with row grains, one-hundred-grain weight, grain yield, and WUE in both years (Table 7). The ear rows were highly significantly positively correlated with grain yield, WUEg, and N partial factor productivity. The row grains showed significant positive correlations with WUEb in both years. The 100-grain weight was positively correlated with grain yield, WUEb, and N partial factor productivity in both years. In addition, the grain yield was correlated with the harvest index, WUEb, WUEg, and N partial factor productivity. In conclusion, the increase in dry matter in aboveground maize plant parts was favorable for an increase in row numbers, 100-grain weight, grain yield, WUEb, and N partial factor productivity, while the increased number of ear rows was favorable for grain yield, WUEg and N partial factor productivity. The increase of ETa reduced WUEg, whereas the increase in dry matter was not conducive to the increase in the harvest index in 2016.

**Table 6.** Effects of the different treatments on grain yield (GY), harvest index (HI), water use efficiency (WUEb and WUEg), and partial factor productivity (PFP).

| Year (Y) | Treatments | | GY (kg ha$^{-1}$) | HI (%) | WUEb (kg ha$^{-1}$ mm$^{-1}$) | WUEg (kg ha$^{-1}$ mm$^{-1}$) | PFP (kg kg$^{-1}$) |
|---|---|---|---|---|---|---|---|
| 2016 | RF | N1 | 5367.67 [d] | 45.43 [b] | 26.35 [c] | 19.44 [b] | 29.82 [d] |
| | | N2 | 6343.84 [b] | 53.09 [a] | 27.40 [bc] | 22.22 [a] | 35.24 [b] |
| | | N3 | 5751.92 [c] | 48.73 [b] | 26.65 [c] | 19.26 [b] | 31.96 [c] |
| | | N4 | 6779.28 [a] | 45.14 [b] | 32.90 [a] | 23.17 [a] | 37.66 [a] |
| | | N5 | 5104.33 [d] | 37.91 [c] | 29.51 [b] | 19.13 [b] | 28.36 [d] |
| | FP | N1 | 4953.56 [c] | 41.60 [b] | 26.78 [b] | 20.45 [b] | 27.52 [c] |
| | | N2 | 5384.80 [b] | 55.71 [a] | 21.81 [c] | 22.04 [a] | 29.92 [b] |
| | | N3 | 5063.21 [c] | 44.60 [b] | 24.74 [b] | 21.40 [a] | 28.13 [c] |
| | | N4 | 5613.71 [a] | 38.41 [b] | 32.84 [a] | 21.95 [a] | 31.19 [a] |
| | | N5 | 4564.73 [d] | 39.87 [b] | 25.64 [b] | 18.99 [c] | 25.36 [d] |
| 2017 | RF | N1 | 4789.29 [c] | 32.51 [a] | 32.85 [d] | 10.68 [b] | 26.61 [c] |
| | | N2 | 5531.85 [b] | 32.09 [a] | 39.48 [c] | 12.65 [a] | 30.73 [b] |
| | | N3 | 4401.93 [d] | 27.32 [c] | 36.42 [b] | 9.94 [c] | 24.46 [d] |
| | | N4 | 5990.22 [a] | 31.26 [ab] | 41.86 [a] | 13.08 [a] | 33.28 [a] |
| | | N5 | 4140.07 [d] | 28.77 [bc] | 31.52 [d] | 9.06 [d] | 23.00 [d] |
| | FP | N1 | 4071.26 [d] | 30.88 [a] | 29.53 [d] | 9.12 [c] | 22.62 [d] |
| | | N2 | 4719.90 [b] | 32.28 [a] | 32.81 [b] | 10.59 [b] | 26.22 [b] |
| | | N3 | 4352.35 [c] | 30.75 [a] | 30.87 [c] | 9.49 [c] | 24.18 [c] |
| | | N4 | 5009.06 [a] | 32.40 [a] | 34.71 [a] | 11.25 [a] | 27.83 [a] |
| | | N5 | 3812.38 [e] | 27.41 [b] | 31.04 [c] | 8.50 [d] | 21.18 [e] |
| CP | | | *** | ns | *** | ** | *** |
| N | | | *** | *** | *** | *** | *** |
| CP × N | | | *** | ns | *** | ** | *** |

Note: GY, grain yield; HI, harvest index; WUEb, water use efficiency of biomass; WUEg, water use efficiency of grain yield; PFP, N partial factor productivity. Y, year, CP, cultivation practice (FP or RF), N, nitrogen formula, with significance indicated as ** for $p < 0.01$, *** for $p < 0.001$; and ns, not significant ($p > 0.05$). Different lowercase letters after the values in the same column indicate that the difference was significant at $p < 0.05$ under LSD test.

**Table 7.** Correlation coefficients between growth and yield parameters of the maize.

| | Item | DM | ETa | Ear Rows | Row Grains | 100-Grain Weight | GY | HI | WUEb | WUEg | PFP |
|---|---|---|---|---|---|---|---|---|---|---|---|
| | | | | | | **2016** | | | | | |
| | **DM** | 1 | 0.344 | 0.309 | 0.475 ** | 0.448 * | 0.435 * | −0.612 ** | 0.987 ** | 0.229 | 0.435 * |
| | **ETa** | 0.004 | 1 | 0.179 | 0.402 * | 0.468 ** | 0.713 ** | 0.255 | 0.377 * | −0.057 | 0.713 ** |
| | **Ear rows** | 0.896 ** | −0.003 | 1 | 0.253 | 0.300 | 0.564 ** | 0.250 | 0.319 | 0.618 ** | 0.564 ** |
| | **Row grains** | 0.878 ** | −0.207 | 0.818 ** | 1 | 0.670 ** | 0.303 | −0.221 | 0.458 * | −0.022 | 0.303 |
| **2017** | **100-Grain Weight** | 0.437 * | 0.205 | 0.360 | 0.408 * | 1 | 0.475 ** | −0.043 | 0.379 * | 0.149 | 0.475 ** |
| | **GY** | 0.860 ** | −0.025 | 0.812 ** | 0.807 ** | 0.389 * | 1 | 0.431 * | 0.446 * | 0.657 ** | 1.000 ** |
| | **HI** | 0.109 | −0.058 | 0.193 | 0.213 | 0.080 | 0.600 ** | 1 | −0.591 ** | 0.364 * | 0.431 * |
| | **WUEb** | 0.982 ** | −0.183 | 0.879 ** | 0.902 ** | 0.388 * | 0.849 ** | 0.118 | 1 | 0.211 | 0.446 * |
| | **WUEg** | 0.842 ** | −0.173 | 0.794 ** | 0.823 ** | 0.348 | 0.989 ** | 0.606 ** | 0.861 ** | 1 | 0.657 ** |
| | **PFP** | 0.860 ** | −0.025 | 0.812 ** | 0.807 ** | 0.389 * | 1.000 ** | 0.600 ** | 0.849 ** | .0989 ** | 1 |

DM: dry matter; ETa: evapotranspiration; GY: grain yield; HI: harvest index; WUEb: water use efficiency of biomass yield; WUEg: water use efficiency of grain yield; PFP: partial factor productivity. * significant at $p < 0.05$, ** significant at $p < 0.01$.

## 4. Discussion

A lack of water is the main limiting factor for crop growth and production in rain-fed areas, and the water stored in the topsoil is easily evaporated. Plastic film mulching can significantly improve the soil water in rain-fed areas and enhance water infiltration [38,39]. In our study, RF increased soil moisture by collecting light rain, promoting rainfall infiltration. A similar result was reported by Thidar et al. [40] who reported that RF helped channel rainfall to the maize root zones effectively and efficiently. In addition, ridge–furrow plastic film mulching reduced overground wind speed, inhibiting evaporation.

RF improved soil moisture and temperature, which promoted the conversion of water from soil evaporation to plant transpiration, increasing the yield of the biomass and grain of maize [41]. In the current study, RF treatments significantly increased the dry matter accumulation compared with FP over the 2016 and 2017 maize growing seasons, likely because the RF treatment had the best soil moisture storage, prolonged the period of crop water availability, and enhanced the crop growth [42]. The mixed $NO_3^-/NH_4^+$ application promotes plant growth by conserving chemical energy and absorbing more carbon [10]. The results of this study support this relationship. $NO_3^-/NH_4^+$ ratios with 3:1 significantly increased plant biomass at physiological maturity (Figure 4). Different $NO_3^-/NH_4^+$ ratios in the N supply influenced the rate of plant growth as well as biomass distribution [43]. This study found that fertilizer with a $NO_3^-/NH_4^+$ ratio of 3:1 (N4) significantly increased the dry matter accumulation of maize at the filling and physiological maturity stages. This outcome was similar to the discovery of Li et al. [27]. The greatest dry matter accumulation was observed with the RFN4 treatment, likely due to it having good hydrothermal and nutrients conditions, which promoted the growth of maize roots and permitted increased water and nutrient uptake [44,45].

The agricultural practice of RF is extensively employed to boost field productivity of most staple crops, such as maize and wheat, and the main reason may be its capacity to raise the soil water content in fleet soil depth [38]. Studies show that alternating large and small RF could significantly improve the soil moisture at depths of 0–200 cm during the early development stage, while increases in soil moisture were decreased in the later stage of maize growth [46]. RF improved the availability of soil water to crops and increased the leaf area index of maize plants with improved soil water utilization efficiency [47]. Due to there being little rainfall during the growing season, our study found that the average soil water content at the 0–200 cm depth was not substantially different between the two examined cultivation strategies in 2017; however, in 2016, RF produced significantly lower soil water content than FP (Figure 2). The relative water content of napier grass (*Pennisetum purpureum schum*.) grown on $NH_4^+$ was found to be lower than that of plants grown on $NO_3^-$ and $NH_4NO_3$ [48]. The utilization of $NO_3^-$-N at an appropriate ratio promoted the photosynthesis of maize leaves after earring and increased the dry matter and grain productive efficiency of water though transpiration [49]. In our study, N4 significantly increased the soil water content of the 0–40 cm soil layer at seedling stage under RF in 2016, while it decreased the soil water content of the 0–40 cm depth in 2017 because the rainfall in the early and middle April of 2016 was significantly higher than that in 2017 (Figure 4).

For rain-fed agriculture, optimized evapotranspiration redistribution strategies were crucial for providing enough soil moisture for maize to prevent acute drought, particularly during water-sensitive periods of the crop [50]. RF decreased ETa compared with flat planting without mulching, regardless of N application [51]. In our research, the cultivation practices had a significant influence on ETa in 2016, but had no significant influence in 2017. This may be related to the annual precipitation because soil moisture maintained balance in the 0–200 cm soil layer when the annual precipitation was over 273 mm under the RF [52]. Suitable $NO_3^-/NH_4^+$ increased the water consumption of maize at different growth stages, and increased the grain yield [6]. In our study, the ETa under N4 ($NO_3^-/NH_4^+$ ratios of 3:1) treatments was higher than those under single $NO_3^-$-N or amidonitrogen ($NO_3^-/NH_4^+$ ratios of 1:0 and urea). A similar result was reported by Cui et al. [6] who compared ETa under different N forms; ETa under $NO_3^-/NH_4^+$ ratios of 3:1 treatments was higher, which was followed by $NO_3^-/NH_4^+$ ratios of 1:0, and 1:3, and $NO_3^-/NH_4^+$ ratios of 1:3 showed the lowest ETa. The highest ETa was observed with the RFN3 treatment in 2016, while with the RFN4 treatment in 2017.

Improvements in the WUE of grain yield have been the focus of researchers and producers [49]. Several previous studies showed that the higher grain yield response of plastic film mulching was mostly attributable to increased topsoil temperature and available soil water content, decreases in the soil evaporation, greater nutrient and water uptake, and lowering of the weed growth [4,46,50]. The ridge–furrow system promoted crop

development and improved grain yield effectively, which was reflected by the increased crop yields (up to 180%) compared to conventional flat planting [53]. In our experiments, the grain yield with RF was 13–14% higher than FP. Different cultivation practices can affect the agronomic traits of a crop [54]. In our research, the higher grain yield of RF was due to the improved ear rows and row grains, while this effect was particularly significant during the drought year 2017.

Maize has a high N requirement, and N levels have a significant impact on grain yield [52]. Both $NH_4^+$-N and $NO_3^-$-N were major forms of mineral N in the soil; however, compared to $NH_4^+$-N, $NO_3^-$-N was generally the principle form [7]. Borgognone et al. [13] proposed that a shift from 100% nitrate to 100% ammonium fertilizer restricted both vegetative growth and fruit yield in tomato plants. N forms significantly affected the grain yield of maize under the same cultivation practices in the current study (Table 4). In both growth seasons of the experiment, N4 (ratio 3:1) treatments produced the highest grain yield of maize, compared with other treatments. The results agree with reports from previous studies by Ochieng et al. [45], showing that $NH_4^+$ treatment, unlike $NO_3^-$ treatment, inhibits the absorption of essential cations and ultimately reduces grain yields. This result is different from Alpha et al. [55] who reported that $NH_4^+$-fed plants produced the highest yield, and this may be due to differences in crops and environmental conditions. Obviously, significant coupling effects of cultivation practices and N forms on maize grain yields were found (Table 4). In the present research, the highest grain yield (6779.28 kg ha$^{-1}$ and 5990.22 kg ha$^{-1}$) of maize was observed for the RFN4 treatment (Table 4).

Maintaining soil moisture and improving WUE was one of the critical factors in the arid and semiarid regions where annual precipitation exceeded evapotranspiration [21]. Consistent with previous studies [56,57], compared with FP, RF significantly improves the WUE of maize in both growth seasons of the experiment; these differences were mainly because RF significantly increased maize grain yields [58]. RF makes better use of the rainwater, thus improving the availability of water to crops, rainwater use efficiency, and grain yields [58]. N4 ($NO_3^-/NH_4^+$ ratios of 3:1) significantly increased the grain yield and WUEb in the two growing seasons (Table 4). This may because a little bit of $NH_4^+$-N could be speedily absorbed by plants and its inhibitory effect was much lower than its nutrient effect [7], and Liu et al. [59] also revealed that mixed $NO_3^-$-N and $NH_4^+$-N improved the wheat growth, chlorophyll content, and $NO_3^-$ reductase activity in leaves compared to either alone. In the present research, the highest WUEb (32.90 kg ha$^{-1}$ mm$^{-1}$ and 41.86 kg ha$^{-1}$ mm$^{-1}$) of maize was observed for the RFN4 treatment (Table 4).

The harvest index had a positive relationship with grain yield and negatively correlated with biological yields [60]. Wang et al. [61] reported that the harvest index under RF was higher by 12.57% than that without film mulching, and the increased harvest index was positively correlated with the grain yield. However, in our study, the difference in harvest index between the two cultivation practices was not significant (Table 5). It was noted that the harvest index was significantly higher with N2 ($NO_3^-/NH_4^+$ ratios of 1:1) treatment than that in the other treatments. N partial factor productivity is regarded as one of the most significant indicators for cereal crops, which directly reflects the economic return of N input into the crop production system [62]. Gu et al. [63] found that continuous ridges with film mulching cultivation increased the PFP of oilseed rape. Our results also demonstrated that the N partial factor productivity under RF was nearly 14% higher than that under FP. This may be due to the higher soil water content under RF, which may have a significant impact on N uptake and utilization [64] because PF might have the ability to improve the N use efficiency by altering water utilization [4]. The N source influenced the N and water use efficiency of maize; the decreasing order was $NH_4NO_3$-N > $NO_3^-$-N > $NH_4^+$-N. Plants cultivated with any of the N forms had their NUE values reduced by $NH_3$ fumigation [65]. There was no difference in N content and N use efficiency between $NH_4^+$ and $NO_3^-$, but the N uptake efficiency of rice with $NH_4^+$ was higher than that with $NO_3^-$ [66]. In our present study, N4 significantly enhanced N partial factor productivity compared with the

other fertilizers, which indicates that the N uptake efficiency of maize was enhanced with the increase in the $NO_3{}^-$-N ratio. Li et al. [28] reported similar findings.

In this study, RF combined with $NO_3{}^-/NH_4{}^+$ ratios of 3:1 reduced water consumption, improved water use efficiency, and increased rainfall infiltration and nitrogen use efficiency, which was favorable to a virtuous cycle of agricultural production. The treatment of RFN4 not only increased the N partial factor productivity but also WUE; both promoted maize development and increased grain yield. This strategy could help to achieve much more efficient use of N fertilizer and to enhance the utilization ratio of water resources, thereby boosting the productivity of dry-land spring maize in rain-fed semiarid environments.

## 5. Conclusions

The ridge–furrow plastic film mulching practice could increase the soil moisture for maize, which could provide favorable conditions for the yield component especially ear rows and row grains, and thus increasing grain yield, WUE, and N partial factor productivity. Compared with other treatments, treatment applying N fertilizer and the $NO_3{}^-/NH_4{}^+$ ratios of 3:1 could result in the following advantages: the nutrient supply could meet the nutrient demands of maize, could promote the dry matter accumulation, could significantly increase the soil water content of the 0–40 cm soil depth when precipitation is over 252 mm, and could increase the grain yield, WUE, and N partial factor productivity. RF combined with the N4 treatment could improve maize yield, WUEb, and N partial factor productivity, which reached the maximum values of 6779.28 and 5990.22 kg ha$^{-1}$, 32.90 and 41.86 kg ha$^{-1}$ mm$^{-1}$, and 37.66 and 33.28 kg kg$^{-1}$, respectively. Hence, we conclude that ridge–furrow plastic film mulching combined with $NO_3{}^-/NH_4{}^+$ ratios of 3:1 could be a better field management for spring maize productivity systems in rain-fed semiarid environment.

**Supplementary Materials:** The supporting information can be downloaded at: https://www.mdpi.com/article/10.3390/agronomy12122943/s1.

**Author Contributions:** Y.G. and L.G. conceived the idea and Y.G. conducted the experiments. H.L. and G.L. helped in the experiments and data collection. Z.C. analyzed the data and prepared the first draft. H.W., Y.W. (Yingze Wang), and Y.W. (Yifan Wang) edited the manuscript. B.W. supervised and contributed to the finalized version and B.Y. proofread the manuscript. All authors have read and agreed to the published version of the manuscript.

**Funding:** This research was funded by the State Key Laboratory of Arid land Crop Science (GSCS-2020-Z6), the Fuxi Outstanding Talent Cultivation Plan of Gansu Agriculture University (Gaufx-02J05), and the Research and demonstration Program on the Optimization of the Construction Path of Maize Industry System and Quality and Efficiency Enhancement Technology for Productive Cultivation in Baiyin City (2022-2-9N).

**Institutional Review Board Statement:** Not applicable.

**Informed Consent Statement:** Not applicable.

**Data Availability Statement:** All data generated or analyzed during this study are included in this published article. The authors declare that all the data supporting the survey result of this manuscript are available within the Supplementary Materials.

**Acknowledgments:** Authors would like to thank those who provided valuable comments and suggestions for the improvement of the manuscript.

**Conflicts of Interest:** The authors declare that they have no known competing financial interest in this paper.

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
