# Peer review of "Optimal Effects of Combined Application of Nitrate and Ammonium Nitrogen Fertilizers with a Ratio of 3:1 on Grain Yield and Water Use Efficiency of Maize Sowed in Ridge–Furrow Plastic Film Mulching in Northwest China"

_agronomy, doi:10.3390/agronomy12122943_

Round 1
Reviewer 1 Report
Some suggestions for improvements were inserted in the text in the form of comments

Author Response
- Abstract: Are these values high, medium or low for the region? What is the average productivity of the region?
Response: These values high for the region.
The average grain yield of maize were 5601 kg ha-1 and 5540 kg ha-1 in 2016 and 2017, respectively. In current study, the grain yield under RFN4 were 6779.28 kg ha-1 and 5990.22 kg ha-1 in 2016 and 2017, respectively. Through consulting a great deal of literature, the author found that the partial factor productivity of maize were 27.0-33.6 kg kg-1 and 24.5-33.3 kg kg-1 in 2016 and 2017, respectively. The water use efficiency were 15.4-18.7 kg ha-1 mm-1 and 6.5-8.7 kg ha-1 mm-1, respectively. The average productivity was greatly affected by rainfall.
L387-L388, Added the sentence “These values were higher than the average values of the region”.
- What is the experimental design you are using? How many repetitions were made of each treatment?
Response: The author added the experimental design and repetitions.
L166-167, The experiment was a two-factor randomized block design with three replications.
- What is your justification or explanation for these changes in plant height and stem diameter?
Response: RF improved root growth and led to a distinct effect on grain yield as compared with FP. In addition, RF reduced soil evaporation, improved dry matter accumulation, distribution, and transpiration rate (Thidar et al., 2020). Mixed supply of nitrate and ammonium fertilizer significantly promoted maize growth and development, improved leaf area and photosynthetic potential (Li et al., 2017). An optimum NO3-/NH4+ ratio can improve plant growth and nutrient uptake efficiency, which increased leaf area, specific leaf weight and total root length (Wang et al., 2019). All of these factors synergistic effect on maize growth and development, and may be the reason for these changes in plant height and stem diameter.
- Remove the three zeros and leave only the whole number. Swap commas for dots in columns.
Response: The author has replaced Figure 4.
L1277, Figure 4, the author Remove the three zeros and leave only the whole number.
Reviewer 2 Report
A lot of research and interesting analyzes have been done. In the methodology, the authors do not refer to standards. Only for works by other authors. Maybe there are no labeling standards in China. The discussion is poorly written. It needs to be changed. Conclusion are badly spelled. These are generalities. There are no recommendations for practice. Written on the basis of the results obtained. 30 comments are highlighted in the text.

Author Response
- Institution: There are missing e-mail addresses.
Response: We added each author's e-mail addresses.
- 2. Abstract: Abstract is too long.
Response: The author has re-condensed the abstractt, and the specific modification as follows:
Abstract: Increasing water use efficiency is important for sustainable agricultural development, especially in arid and semi-arid areas. A two-year field experiments were conducted to investigate the effects of ridge-furrow plastic film mulching (RF) and flat planting plastic film mulching (FP) combined with five different nitrogen (N) fertilizers N1 (KNO3), the nitrate (NO3-)/ammonium (NH4+) mixtures with different pure nitrogen ratios N2 (1:1), N3 (1:3), and N4 (3:1), and the control N5 (urea)) on maize dry matter accumulation, soil water content, grain yield, water use efficiency (WUE) and N partial factor productivity. Our results showed that RF and N4 was more efficient than FP for improving maize grain yield, WUE and nitrogen partial factor productivity, there was a significant interaction for cultivation practices × N formulation. RF and 3:1 NO3-/NH4+ significantly increased grain yield by 14.73%,13.15% and 20.07%, 24.14% in 2016 and 2017, respectively, compared to FP and nitrate only. in 2016 and 2017. Compared to , increased grain yield by . RFN4 produced the highest grain yield in 2016 and 2017 due to the highest dry matter accumulation at filling and physiological maturity stage, ear rows per spike, and row grains per row. Over two growing seasons, WUE and N partial factor productivity under RFN4 was 18.75% and 29.17% on average more than that of other treatments. Therefore RFN4 is an effective cropping system for improving the synchronization of grain yield and water use efficiency for maize production in rain-fed farming.
- 3. Introduction: The brackets are missing.
Response: The author examined the whole manuscript, and added the missing brackets.
L51, [6]; L83, [20]; L504, [21].
- 4. Introduction: It should be 41.26, not 41.264.
Response: Change 41.264 to 41.26 (L76).
- 5. Materials and Method: 2.1 Not “d”only days, review anywhere.
Response: Change “d” to “days” (L121).
- 6. Materials and Method: 2.2 With a dot, revise anywhere.
Response: The author examined the whole manuscript, and added dot in L117, L131, L137, L180, L191, L202, L212, L222, L229, L251, L285, L332, L348, L370, L396.
- 7. Materials and Method: Table 1
Response: Total N, g N kg-1; Available P, mg P kg-1; Available K, mg K kg-1.
- 8. Materials and Method: 2.3 No hm only, ha hectare, revise anywhere.
Response: The author examined the whole manuscript, and modified hm-2 to ha-1.
L143 and L154.
- 9. L159, space
Response: The author adding the “space”, L158, 70 cm wide.
- 10. L166 and L167, 67500? plant per hectare.
Response: Modified “with a density” to “and keep a full stand of seedlings” of 67,500 plants per hectare. L165 and L166.
- 11. Why such a methodology. Why is the 75℃?
Response: 105℃ for 30 minutes is to stop the reaction in the plant, prevent the decomposition of the active substance or other components, in order to determine the content of a compound. The drying process of 60-85℃ has been used, and 75℃ was used in current study.
The author cited literature, L188.
- 12. The letters a, b...in the superscript. Revise anywhere.
Response: The author examined the whole manuscript, and modified letters a, b... in the superscript.
Table 3, Table 5, and Table 6.
- 13. Figure 4, why so many zeros after the decimal point.
Response: The author has replaced Figure 4.
L1277, Figure 4, the author Remove the three zeros and leave only the whole number.
- 14. Discussion: L416, This sentence is not needed in the discussion. This is general.L425, This sentence is not needed in the discussion. This is general. This is not a discussion of results.
Response: The author made the following changes.
L415-L422, Water shortage is the major constraint for crop growth and production in arid and semiarid regions, and water retained in the topsoil is easily lost by evaporation. Plastic film mulching can significantly improve soil moisture in semi-arid regions and enhancing water infiltration [38,39]. In our study, RF increased soil moisture by collecting light rain, promoting rainfall infiltration. A similar result was reported by Thidar et al. [40] who reported that RF helped channel rainfall to the maize root zones effectively and efficiently. In addition, ridge-furrow plastic film mulching reduced over-ground wind speed, inhibited evaporation.
- 15. Discussion: L429, This fragment may be in the introduction not in the discussion. It doesn't have a discussion of the results.
Response: The author deleted this sentence
- 16. Discussion: L432-L435, It doesn't have a discussion of the results. This is not a discussion. This is general statement..
Response: The author made the following changes.
L423-L425, Ridge-furrow plastic film mulching could enhanced soil hydrothermal conditions, promoted the conversion of water from soil evaporation to plant transpiration, increasing the dry-matter and grain yield of maize [41].
- 17. Discussion: L447, This applies to other plants. You don;t have to write about it .
Response: The author deleted this sentence
- 18. Discussion: L454, This sentence is not discussion.
Response: The author made the following changes.
L439-L441, The farming practice of alternating ridges–furrows with plastic film mulching is widely used to improve field productivity of most staple crops such as maize and wheat, mainly because of its ability to increase the soil water content in shallow soil depth [38].
- 19. Discussion: L562, the discussion is not written correctly. It needs to be corrected.
Response: The author made the following changes.
L538-L545, In the study, ridge-furrow plastic film mulching combined with NO3-/NH4+ ratios of 3:1 reduce water consumption, improve water use efficiency, increase rainfall infiltration and nitrogen use efficiency, which was favorable to a virtuous cycle of agricultural production. The treatment of RFN4 not only increased the N partial factor productivity but also WUE, both promoted maize development and increased grain yield. This strategy could help to achieve much more efficient use of N fertilizer and to enhance the utilization ratio of water resources, thereby boosting the productivity of dry-land spring maize in rain-fed semiarid environment.
- 19. Discussion: L577, You shuold write what is the best fertilization. Each nitrogen fertilization increse the yield. Give me a shorter sentence. These are too long and imprecise.
Response: The author made the following changes.
L557-L560, Hence, we conclude that ridge-furrow plastic film mulching combined with NO3-/NH4+ ratios of 3:1 could be a better field management for spring maize productivity systems in rain-fed semiarid environment.

Round 2
Reviewer 2 Report
A lot of research and interesting analyzes have been done. In the methodology, the authors do not refer to standards. Only for works by other authors. Maybe there are no labeling standards in China. The discussion and conclusions have been corrected. Many comments have been corrected. There are 3 small bugs left. Marked in the text.
